# NMR Characterization of Long-Chain Fatty Acylcarnitine Binding to the Mitochondrial Carnitine/Acylcarnitine Carrier

**DOI:** 10.3390/ijms23094608

**Published:** 2022-04-21

**Authors:** Ningning Zhang, Xiaopu Jia, Shuai Fan, Bin Wu, Shuqing Wang, Bo OuYang

**Affiliations:** 1State Key Laboratory of Molecular Biology, Shanghai Institute of Biochemistry and Cell Biology, Center for Excellence in Molecular Cell Science, Chinese Academy of Sciences, Shanghai 200031, China; zhangningning2014@sibcb.ac.cn; 2University of Chinese Academy of Sciences, Beijing 100049, China; 3School of Pharmacy, Tianjin Medical University, Tianjin 300070, China; 20216020604@tmu.edu.cn (X.J.); 20216020602@tmu.edu.cn (S.F.); 4National Facility for Protein Science in Shanghai, ZhangJiang Laboratory, Shanghai Advanced Research Institute, Chinese Academy of Sciences, Shanghai 201203, China; bin.wu@sibcb.ac.cn

**Keywords:** CAC, long-chain acylcarnitine, cytosol-open state, matrix-open state, NMR, molecular dynamics simulation

## Abstract

The mitochondrial carnitine/acylcarnitine carrier (CAC) transports short-, medium- and long-carbon chain acylcarnitines across the mitochondrial inner membrane in exchange for carnitine. How CAC recognizes the substrates with various fatty acyl groups, especially long-chain fatty acyl groups, remains unclear. Here, using nuclear magnetic resonance (NMR) technology, we have shown that the CAC protein reconstituted into a micelle system exhibits a typical six transmembrane structure of the mitochondrial carrier family. The chemical shift perturbation patterns of different fatty acylcarnitines suggested that the segment A76–G81 in CAC specifically responds to the long-chain fatty acylcarnitine. Molecular dynamics (MD) simulations of palmitoyl-L-carnitine inside the CAC channel showed the respective interaction and motion of the long-chain acylcarnitine in CAC at the cytosol-open state and matrix-open state. Our data provided a molecular-based understanding of CAC structure and transport mechanism.

## 1. Introduction

The degradation metabolism of fatty acid primarily takes place in the mitochondrial matrix. The short-chain fatty acids can directly pass through the mitochondrial membrane, while the medium- and long- chain fatty acids require a transport system to move across the inner mitochondrial membrane. A small molecule, carnitine, serves in transporting medium- or long- chain fatty acids across the mitochondrial membrane, by forming acylcarnitines. The mitochondrial carnitine/acylcarnitine carrier (CAC) is the transport protein that delivers fatty acylcarnitines from the cytosol into the mitochondrial matrix in exchange for intramitochondrial free carnitines [1,2]. The deficiency of CAC causes severe diseases, such as hypoglycemia, hypoketosis, hyperammonemia and damages in multiple organs [2]. CAC has great potential as a target for therapeutic intervention and CAC-deficient patients with mild phenotypes have been treated using statins and/or fibrates as a pharmacological approach in clinical studies [3].

CAC belongs to a mitochondrial carrier family (MCF) that is localized in the mitochondrial inner membrane and mediates the passage of a variety of molecules between the matrix and intermembrane space [4]. Generally, MCF proteins consist of six transmembrane α-helices and three short matrix α-helices, which can be divided into three tandemly repeated homologous domains [5]. ADP/ATP carrier (AAC) is the most well studied mitochondrial carrier. The crystal structures of AAC showed that AAC has six transmembrane helices (TMH), as predicted, and gives a three-fold symmetry [6,7]. The residues in the middle of TMH can form salt bridges, locking AAC at two distinct conformation states, matrix-open (m-state) or cytosol-open (c-state) [8]. These salt bridges are thought of as the substrate binding sites [9]. On the other hand, substrates are the modulators of the salt bridges, leading to the rearrangement and conformation switch of AAC [10]. This common substrate binding site model may explain the substrate recognition mechanism of many mitochondrial carriers with charged substrates, such as ions and nucleotides. However, it is hard to explain how CAC recognizes fatty acylcarnitines, since the substrate of CAC has a long hydrophobic fatty acyl group in addition to the charged headgroups. Moreover, CAC can transport acylcarnitines with different fatty acid chain lengths, and the detailed mechanism of the selectivity of fatty acyl groups remains unclear.

Previously, Indiveri and Palmieri attempted to address the CAC substrate-binding question using the site-directed mutagenesis approach and liposome transporting experiments [11]. Based on the AAC crystal structures (PDB code: 1OKC) [6], they built a homology model of CAC and proposed several hydrophobic residues as the potential fatty acyl group binding sites. V25, P78, V82, M84 and C89 in the first and second transmembrane α-helices were reported to bind the carbon chain of the acylcarnitines through hydrophobic interactions, while a highly conserved H29 forms an H-bond with the substrate to facilitate the correct positioning of the substrate. The impairment or loss of function observed in the CAC mutants demonstrated that R178, D179 and R275 are also crucial residues involved in carnitine binding and translocation [12]. However, the structural visualization of the substrate recognition is still lacking.

Here, we used nuclear magnetic resonance (NMR) technology to investigate the structural basis of the substrate recognition in CAC. Backbone assignments of CAC protein in a micelle system revealed CAC exhibits a typical six TMH structure of the mitochondrial carrier family. NMR titration experiments using carnitine and different fatty acylcarnitines showed that the segment A76–G81 in CAC specifically responds to palmitoyl-L-carnitine, which is consistent with the previous results that P78 plays an important role in the transport activity and G81 is a pathogenic site [13]. Molecular dynamics (MD) simulations for palmitoyl-L-carnitine and CAC showed that palmitoyl-L-carnitine interacts with the A76–G81 region more closely at the cytosol-open state and moves in the CAC passageway down to the CAC matrix gate. Therefore, our data confirmed the hydrophobic substrate binding pocket reported previously and visualized the motion of palmitoyl-L-carnitine inside CAC, which is potentially useful for mechanistic understanding of the CAC transport mechanism.

## 2. Results

### 2.1. An NMR-Feasible Sample of CAC

The expression of human CAC (residues 1–301) with a C-terminal His_8_ tag in a pET-28a vector was tested in *E. coli* BL21 (DE3) and *E. coli* C41 (DE3). After the induction by isopropyl β-d-1-thiogalactopyranoside (IPTG) at 37 °C for 3.5 h and 5.5 h, the proteins expressed in strain BL21 (DE3) showed higher protein expression levels, which was further selected to express CAC proteins for NMR sample preparation (Figure 1A). The overexpressed CAC proteins were enriched in inclusion bodies; therefore, all the cysteines in the human CAC sequence were mutated into serines to avoid the misfolding during the reconstitution. Following the purification scheme of the mitochondrial transport protein family [14,15], the overexpressed CAC in the inclusion bodies was dissolved by sarkosyl and reconstituted into dodecylphosphocholine (DPC) detergent micelles. The solubilized CAC was then subjected to a series of purification steps, including Ni-NTA affinity and size exclusion chromatography (Figure 1B). The purity of the eluted fractions was analyzed by SDS-PAGE (Figure 1C). The pure fractions were then concentrated to make a final NMR sample containing 0.6 mM CAC, 60 mM DPC, 30 mM 2-(N-morpholino) ethanesulfonic acid (MES) (pH 6.0), and 20 mM NaCl. The two-dimensional (2D) ^1^H-^15^N transverse relaxation optimized spectroscopy-heteronuclear single quantum coherence (TROSY-HSQC) NMR spectrum showed good dispersion and resonance homogeneity (Figure 1D).

### 2.2. Resonance Assignment and Initial Structure Characterization of CAC

Using a combination of triple-resonance experiments and 3D ^15^N-edited nuclear Overhauser effect spectroscopy (NOESY), we assigned about 85% of the backbone resonances of non-proline residues (Figure 2A). The majority of the unassigned residues are in the region from 35 to 51, possibly due to exchange broadening. The backbone chemical shifts were analyzed by the TALOS+ program [16] to derive backbone dihedral angles and the secondary structure of CAC (Figure 2B). Despite the low similarity in their sequences and NMR spectra, the results showed that the secondary structure of CAC is similar to the reported mitochondrial carrier proteins, such as AAC, characteristic of six long TMH with short breaks and three amphipathic helices (Figure 2B). The overall agreement in helical structural arrangements between CAC and AAC suggests that these two homologous carriers resemble each other in their structures.

### 2.3. Substrate Binding to CAC from Chemical-Shift Titrations

To investigate carnitine/acylcarnitine binding sites, we used three different substrates to titrate CAC samples (Appendix A). The addition of carnitine into CAC caused very little chemical shift perturbations, even at a very high carnitine concentration (120 mM) (Appendix A). Palmitoyl-L-carnitine (C16 carnitine) is the major form of long-chain fatty acylcarnitine in the physiological environment. When titrated with palmitoyl-L-carnitine, the statistics of chemical shift perturbation of the assigned signals showed a significant response at A76~G81 (Figure 3B) and I257~V262. However, the region I257~V262 locates in the mitochondrial matrix, which is far from the transmembrane passageway of CAC. Considering the hydrophobic properties of palmitoyl-L-carnitine, A76~G81 is likely to be the region where the fatty acyl group binds.

Decanoyl-L-carnitine (C10 carnitine) was used to explore the recognition for short-chain fatty acylcarnitine. Although decanoyl-L-carnitine also contains a hydrophobic fatty acid group, the NMR spectra of decanoyl-L-carnitine and palmitoyl-L-carnitine showed different perturbations. The statistics of chemical shift changes showed that the response of CAC to decanoyl-L-carnitine was significantly weaker than that of palmitoyl-L-carnitine, and lacked characteristic patterns. Particularly, A77, I79 and G81 did not show strong responses compared to what was observed from palmitoyl-L-carnitine titration (Appendix A). Therefore, it is not clear which region in CAC can bind to decanoyl-L-carnitine specifically. One interpretation for the big differences observed between palmitoyl-L-carnitine and decanoyl-L-carnitine is that they are possibly due to the different length of the fatty acid group.

We further tried to use nuclear Overhauser enhancement (NOE) experiments to confirm the long chain fatty acyl group binding sites. To unambiguously detect the signals between CAC and palmitoyl-L-carnitine in the NOE experiments, CAC was fully deuterated during growth, and deuterated DPC was used in protein purification and reconstitution. However, a large amount of sarkosyl, with similar acyl signals as palmitoyl-L-carnitine, was also used during the purification, which is very difficult to be completely removed through dialysis and hence, caused the failure of the unambiguous NOE signal detection for palmitoyl-L-carnitine. We also tried to label palmitoyl-L-carnitine with a paramagnetic nitroxide group and use PRE experiments to identify the binding sites. However, the high hydrophobicity of the long acyl chain caused trouble in the synthesis and separation of the nitroxide labeled palmitoyl-L-carnitine; no PRE probe was successfully produced to perform the PRE experiment. Nevertheless, based on the response of CAC to different substrates and previous mutagenesis results, the A76~G81 region may be the region mediating the specific recognition of long-chain fatty acylcarnitine.

### 2.4. Molecular Docking of Palmitoyl-L-Carnitine to CAC

To elucidate the binding site in the CAC protein, we first generated two CAC homology models using the AAC crystal structures of 1OKC (PDB code) [6] and 6GCI (PDB code) [8]. The sequence alignment of CAC with AAC showed that they have about 25% sequence similarity (Figure 2B). The crystal structure of 1OKC represents a c-state open conformation, while 6GCI represents an m-state open conformation. Molecular docking was performed using AutoDock Vina1.1.2 [17] embedded in PyRx [18]. The interaction region (A76~G81) from NMR titration experiments was integrated into the docking of palmitoyl-L-carnitine into the CAC models. This A76~G81 region interacts with the long fatty acyl group through hydrophobic contacts, in which A77, P78, I80 and G81 are the key interaction residues (Appendix A). The final docked poses (Appendix A) showed that palmitoyl-L-carnitine interacts more closely with CAC at the A76–G81 region in the c-state open conformation than that in the m-state open conformation. Other amino acids involved in the substrate interaction in the c-state open conformation after MD docking are D32, K35, E132, K135, R178, D231 and R275. They largely interact with the headgroup of palmitoyl-L-carnitine. Significant differences are observed in the m-state open complex after MD docking; the headgroup of palmitoyl-L-carnitine mainly interacts with S89, R178, D179, N223, W224 and R275, indicating the headgroup reorients its direction in the m-state open complex.

### 2.5. The Analysis of the MD Simulations of the Palmitoyl-L-Carnitine with CAC

In an attempt to explore the motion characteristics of palmitoyl-L-carnitine within the two CAC states, the analysis of the 100 ns MD trajectory for each state is depicted. We first extracted the average conformation of each complex at the initial 1 ns and the final 1 ns duration at equilibrium, respectively (Figure 4 and Appendix A), and elucidated the motions of protein and palmitoyl-L-carnitine by calculating the Root Mean Square Deviation (RMSD) values and the Root Mean Square Fluctuation (RMSF) values for CAC and palmitoyl-L-carnitine atoms (Appendix A), where the RMSD is used to measure the average change of the molecular dynamic stability, and the RMSF is useful for characterizing local changes along the protein backbone and molecular atoms. RMSD plots of the apo-protein models showed large RMSD values between the initial state and the final state, but the protein reached an equilibrium after 70 ns (Appendix A). The stable averaged structures (c-state open and m-state open) were selected for the docking mentioned above. The RMSD value of the c-state open complex is smaller than 2.4 Å, while the RMSD value of the m-state open complex is smaller than 1.6 Å, indicating that the m-state open complex is more stable, and the c-state open is undergoing a larger conformational change during the simulation. The RMSD values for the ligand also showed that the ligand moves more freely in the c-state open carrier model. Since the ligand is quite flexible and dynamic, the RMSF values are relatively large, especially for the acyl tail.

After 100 ns MD simulation with palmitoyl-L-carnitine, the final snapshot from the two models was extracted, respectively. The interaction of palmitoyl-L-carnitine with CAC in the c-state open complex involves the residues of L24, V25, D32, E132, K135, D179, R275 and Q280 (Appendix A). Most of them are polar or charged residues. Fewer residues (mainly R275) in CAC contribute strong interactions with palmitoyl-L-carnitine in the m-state open complex (Appendix A). Analysis of the interaction types between the substrate and the protein indicates that R275 interacts with the headgroup of palmitoyl-L-carnitine mainly through hydrogen bonds. Meanwhile, several residues interact with the substrate through water bridges in the c-state open conformation, indicating there are many water molecules around the substrate. Compared with the m-state open conformation, the substrate is more exposed to the solvent.

Interestingly, in both states palmitoyl-L-carnitine tends to shift downward from the initial docking pose to the bottom of CAC after 100 ns MD simulation (Figure 4). The comparison of the initial (Ci) and the final (Cf) c-state open conformations revealed a RMSD value 0.543 Å, but an obvious contraction from Ci to Cf for the mouth of CAC (Figure 4B, Co, the overlap of Ci and Cf), while the final m-state open conformation (Mf) does not show significant changes comparing to the initial m-state open conformation (Mi) (Figure 4D, Mo, the overlap of Mi and Mf), with a smaller RMSD value 0.427 Å (Appendix A). We did not observe a transition from c-state to m-state of CAC after the 100 ns c-state MD simulation, from which we infer that the 100 ns timescale only simulates the beginning of the transition.

## 3. Discussion

In this study, we have investigated CAC in DPC micelles using NMR technology. Detergent micelles are the most common membrane-mimetic media used in solution NMR to solubilize membrane proteins. However, the use of DPC is in debate, since DPC is a relatively strong detergent that performed poorly in preserving the activities of membrane proteins in some cases [19]. Lipid bicelles or nanodiscs that more closely resemble the natural lipid bilayers have been developed to study membrane proteins. We previously tried to incorporate mitochondrial carriers, such as AAC, uncoupling protein 1 (UCP1), into the lipid bilayer systems, but failed to obtain usable NMR spectra due to the large size and high hydrophobicity of these carriers. Instead, our earlier studies showed that DPC-reconstituted AAC can preserve the binding of cardiolipin, a functionally important lipid to AAC [20]. Meanwhile, the fatty acid binding site of UCP1 [14] and the MgATP binding site of the short Ca^2+^-binding mitochondrial carrier (SCaMC) [15] identified by NMR methods in DPC micelles were verified by functional mutagenesis using liposome assays. These results indicate that DPC detergent micelle is a suitable system to study mitochondrial carriers. Therefore, the choice of DPC micelles to characterize CAC here is a viable solution.

Our NMR data recorded in DPC micelles revealed that the helical segments of CAC agree well with those of AAC, indicating CAC has a similar overall architecture to that of AAC. Through NMR titration experiments, we found carnitine alone did not cause significant chemical shift perturbations on the CAC spectrum. Decanoyl-L-carnitine caused stronger chemical shift perturbations than carnitine, but it is hard to identify the specific interacting regions. Palmitoyl-L-carnitine derived from 16-carbon long-chain fatty acids showed obvious chemical shift perturbations, and A76–G81 specifically respond to palmitoyl-L-carnitine. This substrate binding site is consistent with previous mutagenesis work suggesting that the second transmembrane helix is the major region for recognizing the tail of long-chain fatty acids [21]. Mutating these hydrophobic residues disrupted the transport activity of CAC for multiple substrates, possibly by disturbing the interaction between CAC and the substrates, or the conformation of CAC itself. It is worthy to note that the chemical shift perturbations observed by our NMR experiments may also originate from the disturbance of the membrane by palmitoyl-L-carnitine. Therefore, further investigations to clarify the substrate recognition mechanism are required.

In the previous site-directed mutagenesis experiments, both P78A and P78M significantly reduced the transport activity of long-chain fatty acylcarnitine [21]. Since Pro does not contain a backbone amide proton, the signal of Pro cannot be detected on the ^1^H-^15^N TROSY-HSQC NMR experiment. However, the adjacent A76, A77 and I79 were significantly affected by palmitoyl-L-carnitine, indicating that P78 is an important site for the recognition of the hydrophobic tail of long-chain fatty acylcarnitine. G81R is one of the important pathogenic mutations [13]. Previous studies suggested that G81 is involved in substrate recognition, and mutation of G81 may interfere with substrate binding activity and hence lead to CAC deficiency related diseases, which is consistent with our NMR titration result that G81 has an obvious response to long-chain fatty acylcarnitine. The key residues observed in the MD simulations shifted away from the NMR identified region and only P78 maintained a weak hydrophobic interaction with the substrate in the c-state open complex. However, strong interacting residues from MD simulations, including V25, E132, D179 and R275, were also verified in earlier studies that marked changes in the transport activity of CAC were reported when replacing these residues individually with alanine [12,21]. These results indicate that multiple residues along the channel passage facilitate the delivery of the substrate from the cytosol to the mitochondrial matrix.

MD simulations illustrated the molecular interaction and motion for palmitoyl-L-carnitine inside the CAC channel. Our MD results suggested that palmitoyl-L-carnitine interacts with several polar or charged residues in CAC, and moves downward to the bottom of CAC to release into the mitochondrial matrix. R275 is the major interaction residue in both complexes. In a previous MD simulation of AAC [22], R279, the aligned residue in AAC, has been reported to be the essential binding residue of the phosphate group of ADP, which forms a salt bridge with E29 and plays an important role in the conformational transition. The involvement of the same charged residue in both carriers indicates that the electrostatic network of mitochondrial carriers is critical in transport activity. No transition from the c-state open to the m-state open was detected in the simulation within our short simulation time. The c-state open AAC also showed no conformational transition to the m-state open after a 1.5 μs simulation [23], suggesting that a longer period of MD simulation is required to observe the transition between the two states of mitochondrial carriers. Up to now, the transition time of the complex from the c-state open to the m-state open through dynamics simulations is unknown. More research is needed to fully understand the transport conformational switch.

## 4. Materials and Methods

### 4.1. Protein Expression and Purification

Human gene CAC in which all Cys are mutated into Ser with a C-terminal His8 tag was cloned into the pET28a expression vector. The plasmid containing the CAC insertion was transformed into the *E. coli* BL21 (DE3) (New England BioLabs, Ipswich, MA, USA) expression strain. The cells were grown at 37 °C in M9 minimal medium until OD_600_ reached 0.8–1.0. After induction with 0.5 mM IPTG, the protein was expressed for 4 h at 37 °C. CAC was expressed in inclusion bodies and refolded according to a previously established protocol, to generate the folded protein for functional assays [24].

The cells were lysed in buffer containing 50 mM Tris, pH 8.0 and 150 mM NaCl. After centrifugation at 20,000× *g* for 30 min, the pellet containing the inclusion body was collected and solubilized with buffer containing 1.5% (*w/v*) N-lauroylsarcosine (sarkosyl), 50 mM Tris, pH 8.0 and 150 mM NaCl. After solubilization at 4 °C for 16 h, the solution was centrifuged at 20,000× *g* for 30 min to remove the insoluble debris. The supernatant containing the solubilized CAC was rapidly diluted three-fold with buffer containing 6 mM dodecylphosphocholine (DPC), 50 mM Tris, pH 8.0, 150 mM NaCl and 20 mM imidazole. The diluted solution was then passed through Ni-NTA resin (Cytiva, Marlborough, MA, USA), pre-equilibrated with dilution buffer. The protein was washed in buffer containing 6 mM dodecylphosphocholine (DPC), 50 mM Tris, pH 8.0, 150 mM NaCl and 30 mM imidazole, and then eluted in the same buffer with 500 mM imidazole. The elute was passed through a Superdex 200 size-exclusion column (Cytiva, Marlborough, MA, USA) in FPLC buffer containing 30 mM MES, pH 6.0, 20 mM NaCl and 3 mM DPC. The fractions in the major peak were collected, and passed through a Superdex 200 size-exclusion column again in the same buffer. The homogenous CAC fractions in the secondary size exclusion chromatography were pooled and concentrated to produce the final NMR sample containing 0.6 mM CAC, 60 mM DPC, 30 mM MES, pH 6.0 and 20 mM NaCl.

### 4.2. NMR Resonance Assignment

Sequence-specific assignment of backbone ^1^H^N^, ^15^N, ^13^Cα, ^13^Cβ and ^13^C′ resonances were accomplished using 3D TROSY-based HNCA, HN(CO)CA, HNCACB, HN(CA)CO and HNCO experiments [25,26]. Additionally, the assignments were validated using a 3D (H^N^, H^N^)-HSQC-NOESY-TROSY spectrum. These experiments were performed using a 0.6 mM *U*-[^15^N, ^13^C, ^2^H] CAC sample at 33 °C on a 600 MHz Bruker spectrometer equipped with a cryogenic probe. The NMR spectra were processed using NMRPipe [27] and analyzed using XEASY [28] and CcpNmr [29].

### 4.3. NMR Titration Experiments

L-carnitine, palmitoyl-L-carnitine and decanoyl-L-carnitine were dissolved by water to prepare stock solution (L-carnitine: 1 M; palmitoyl-L-carnitine: 50 mM; decanoyl-L-carnitine: 100 mM). For carnitine titration, the concentration gradients were 0 mM, 1 mM, 5 mM, 10 mM, 20 mM, 40 mM, 60 mM and 120 mM. For decanoyl-L-carnitine, the concentration gradients were 0 mM, 0.5 mM, 1.5 mM, 3 mM, 6 mM and 10 mM. For palmitoyl-L-carnitine, the concentration gradients were 0 mM, 0.1 mM, 0.5 mM, 1.5 mM, 3 mM, 6 mM and 10 mM.

### 4.4. Protein Modeling

The CAC protein sequence searched from the the NCBI (www.ncbi.nlm.nih.gov, accessed on 16 March 2016) with the code NP_000378.1 [30] was submitted to the Swiss-Model server online (http://swissmodel.expasy.org, accessed on 11 December 2021). Two 3D crystal structures of c-state and m-state 1OKC and 6GCI were selected as templates to build the CAC models, respectively. These homology models were evaluated using QMEANDisCo Global. Due to the low sequence similarity to AAC, the QMEANDisCo Global score is relatively low with the values 0.59 ± 0.05 for the c-state model (1OKC template) and 0.58 ± 0.05 for the m-state model (6GCI template). In order to improve the quality of the models, we carried out the MD simulation for the protein models without substrate, and the stable averaged protein structures from 70–100 ns were extracted for further docking and MD simulation.

### 4.5. Molecular Docking

The 3D ligand structures of palmitoyl-L-carnitine were prepared using Discovery Studio 4.0 [31]. Its geometry was pre-optimized by molecular mechanics using steepest descent and conjugate gradient. Molecular docking was performed using the Autodock Vina 1.1.2 [17] software embedded in PyRx [18], with default scoring function parameters. The PDB files and ligand were converted to the PDBQT files using PyRx [18]. The grid box size was approximately set to cuboid in order to enclose the entire protein channel. The approximate binding position of the ligand determined by NMR spectra was defined as the center of the binding shell during docking. The side length of the docking square shell was set to be 5 Å longer than the length of the ligand. The ligand searches around the NMR-identified region inside the shell to find the best pose. In addition, the other parameters’ values were set to default. The binding free energy of each protein-ligand was calculated from the built-in algorithm in Autodock Vina 1.1.2 [17]. The best-fitted poses for each model were used for MD simulation.

### 4.6. Molecular Dynamics Simulations

The initial optimized docked complex from the two states was individually imported into Desmond’s system builder module. A membrane bilayer system consisting of 1-palmitoyl-2-oleoyl-sn-glycero-3-phosphocholine (POPC) was automatically added to the protein–ligand complex. The explicit simple point charge (SPC) water molecules were also introduced into the POPC bilayer system. The MD simulations were then performed using the Desmond 2020 package [32], under the force field OPLS2005 [33], in a bilayer with an appropriate number of counter ions to balance the net charge of the system solvated in 0.15 M NaCl. Nose–Hoover temperature coupling [34] and the Martina–Tobias–Klein method [35,36] with isotropic scaling were used to control the simulation temperature (300 K) and atmospheric pressure (1 atm). The particle-mesh Ewald (PME) method [37,38] was used to calculate long-range electrostatic interactions with grid spacing of 0.8 Å. Van der Waals (VDW) and short-range electrostatic interactions were smoothly truncated at 9.0 Å. Before MD simulations, the system was equilibrated using the default membrane relax protocol provided in Desmond [32], which consists of a series of restrained minimizations and a heating process that are designed to slowly relax the system, without deviating too much from the initial protein coordinates. After 2 ns system minimization and relaxation, each system was subject to 100 ns of normal pressure and temperature (NPT) production simulation, during which the configuration was saved every 100 ps.

## 5. Conclusions

In summary, we first developed a CAC sample in a micelle system that can generate high-resolution NMR spectra and thus affords comprehensive characterization of CAC secondary structure and chemical-shift analysis. Our NMR studies of CAC have shown that CAC consists of six transmembrane helices and resembles AAC structure. The titrations of fatty acylcarnitines showed that only palmitoyl-L-carnitine caused specific chemical shift perturbation patterns and that residues A76–G81 in CAC respond significantly to the long-chain fatty acylcarnitine. MD simulations of the palmitoyl-L-carnitine and CAC complex showed a fast and dynamic motion of long-chain fatty acylcarnitine in CAC and that palmitoyl-L-carnitine moves away from the initial binding pose after a short simulation time. In both c-state open and m-state open complexes, R275 is an essential interaction residue. Together, these results enhance our understanding of the transport mechanism of CAC.

## Figures and Tables

**Figure 1 ijms-23-04608-f001:**
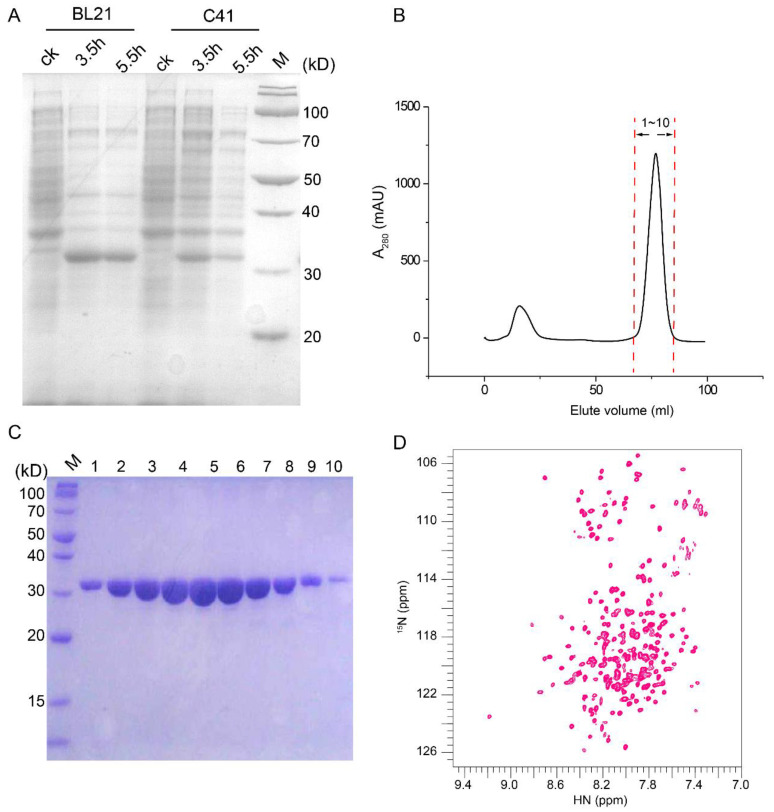
Protein purification and NMR sample preparation of CAC. (**A**) The effects of *E. coli* strains and culture time at the expression level of CAC at 37 °C; (**B**) Size exclusion chromatography, column: HiLoad 16/600 SuperdexTM 200 pg, buffer: 30 mM MES, 20 mM NaCl, 0.1% DPC, pH = 6.0; (**C**) SDS-PAGE analysis of fractions in size exclusion chromatography; (**D**) 2D ^1^H-^15^N TROSY-HSQC NMR spectrum of CAC in DPC micelles.

**Figure 2 ijms-23-04608-f002:**
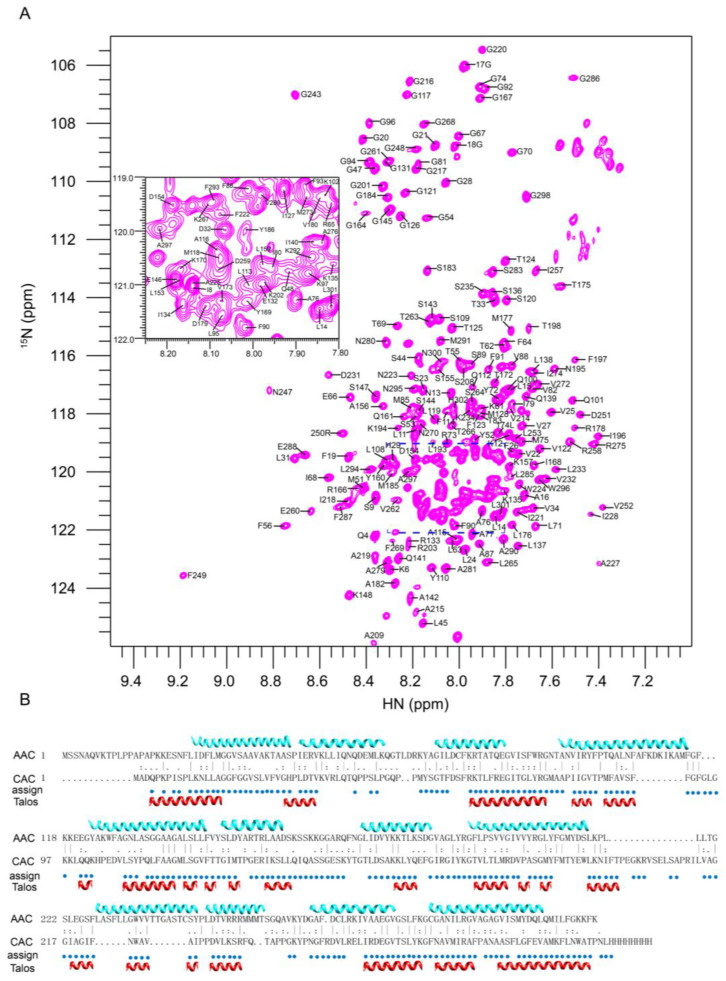
Backbone assignment and secondary structure prediction. (**A**) Backbone assignment of signals in 2D ^1^H-^15^N TROSY-HSQC spectrum; (**B**) Sequence alignment between CAC and AAC, and secondary structure assignment of CAC; cyan: secondary structure assignment of AAC from crystal structure; red: secondary structure assignment of CAC predicted by TALOS+.

**Figure 3 ijms-23-04608-f003:**
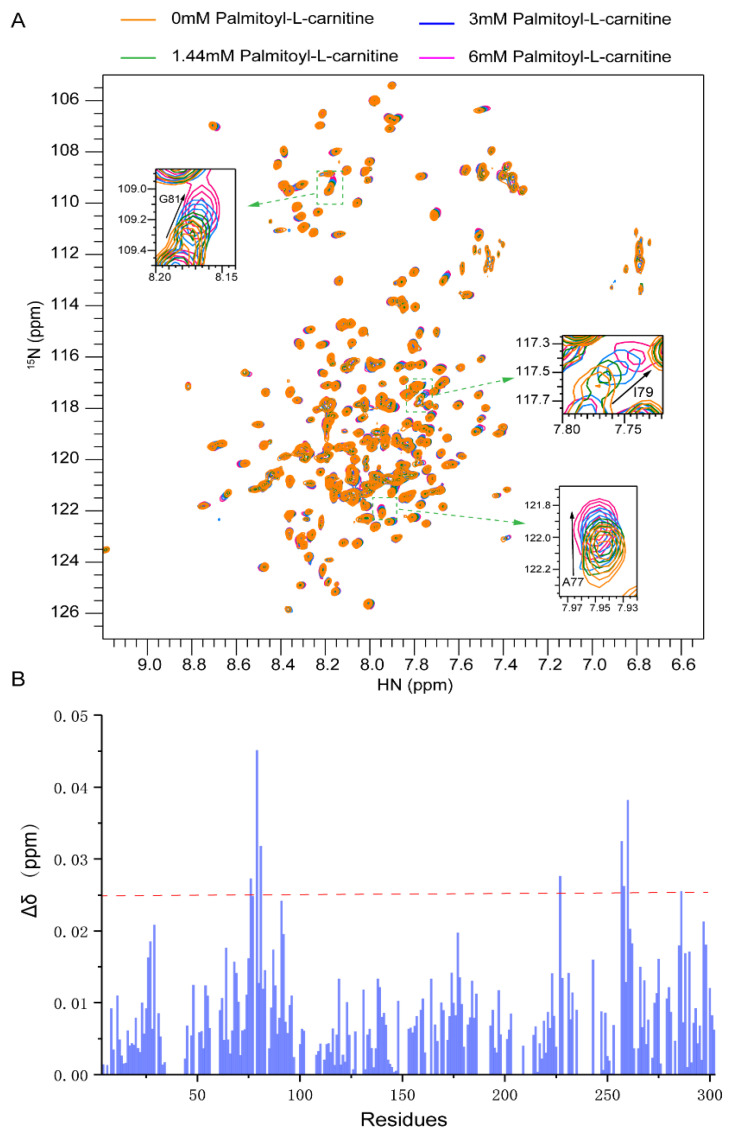
NMR titrations of palmitoyl-L-carnitine into CAC. (**A**) Superimposed 2D ^1^H-^15^N TROSY-HSQC spectra with palmitoyl-L-carnitine in multiple concentrations; (**B**) The residue-specific chemical shift changes of the 2D ^1^H-^15^N TROSY-HSQC spectra between CAC with and without 6 mM palmitoyl-L-carnitine.

**Figure 4 ijms-23-04608-f004:**
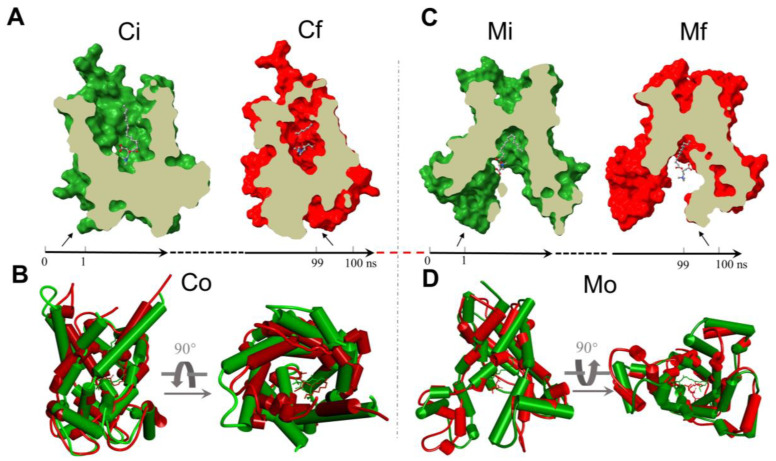
The average structures of the initial 1 ns (green) and final 1 ns (red) within 100 ns MD simulations for c-state open (C) and m-state open (M) CAC proteins, respectively. (**A**) Ci (initial) and Cf (final) indicate the downward translocation of the palmitoyl-L-carnitine within the c-state open CAC channel; (**B**) Co, the overlapped structures of c-state CAC (Ci and Cf), indicating an obvious trend of contraction for the mouth of c-state open; (**C**) Mi (initial) and Mf (final) indicate the motion of the palmitoyl-L-carnitine within the m-state open CAC channel; (**D**) Mo, the overlapped structures of m-state open CAC, indicating the dynamic status of m-state open CAC protein.

## Data Availability

Not applicable.

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
