# Peer review of "NMR Characterization of Long-Chain Fatty Acylcarnitine Binding to the Mitochondrial Carnitine/Acylcarnitine Carrier"

_ijms, 2022, doi:10.3390/ijms23094608_

Round 1

Reviewer 1 Report

The work entitled “NMR characterization of long-chain fatty acylcarnitine binding to the mitochondrial carnitine/acylcarnitine carrier” combines experimental and computational work to investigate the structural basis of the carnitine/acylcarnitine carrier (CAC) substrate recognition. Experimental data were obtained from nuclear magnetic resonance (NMR). Docking and Molecular dynamics (MD) simulations were done for further characterization of the interaction of Palmitoyl-L-carnitine with CAC.

The work is of interest. However, the significance of the data is questionably in several aspects. I feel that there are several questions both from the experimental and from the computational side to be addressed before an eventual indication for publication.

1 - The impact of using a micelle system, instead of a lipid bilayer system in the structure of the CAC protein should be discussed.

2 – Please, improve the resolution of the figures. For instance, Figure 2 is very bad, and it not possible to have an acceptable view of the data.

3 - The choice of the concentrations used in the titration experiments should be justified. Substrate concentrations are very likely to be above the critical aggregation concentration of the substrates, which means that substrates are in the form of aggregates that may interact in a different mode with the protein, than if they were in the form of monomers. Indeed, this issue seems to be briefly mentioned in the discussion (“It is worthy to note that the chemical shift perturbations observed by our NMR experiments may also come from the disturbance of the membrane by palmitoyl-L-carnitine.”), which means that authors may be aware of the problem. Also, justify the physiological meaning of the used concentrations.

4 - Since a titration of a protein with substrates is being made, in theory, it should be possible to extract a binding constant from the observable variable (for instance DeltaSigma) with the substrate concentration. This is relevant and is also an important confirmation about the quality of the obtained data. This is also related with my previous comment, and, if experiments were done correctly, clear curves should be obtained. I suggest this to be done and the data included in the main text.

5 – Please, describe the validation of the initial protein models used in the simulations.

6 – For the MD simulations, the validation of the protein structures without ligand is required to prove that the conditions of the simulations are suitable for this work.

7 - Describe the protocol for insertion of the protein and ligand molecules in the lipid bilayer. Also, include a section describing the analysis of the MD simulations.

8 - Figure 4, namely 4A and 4C is not understandable. However, the fact that the ligand tends to shift from the initial docking binding pose, in relatively short MD simulations, seems to indicate that the docking binding pose is not the true energy minima, causing doubts about the identified binding site. In my opinion this puts more questions than certainties about this work and, should be addressed in the text.

9 – With exception to the reference to the residues A76-G81, conclusions are meaningless.

Reviewer 2 Report

In the manuscript IJMS-1651531, Ningning Zhang et al. have shown by NMR and MD simulations that the region A76-G81 of the carnitine/acylcarnitine carrier (CAC) specifically interacts with palmitoyl-L-carnitine. This manuscript might be accepted for publication provided that the comments reported below are dealt with properly.

  1. The authors have generated two homology models of CAC based on the 3D structures of the ADP/ATP carrier PDB 10KC and 6GCI. These homology models should be analyzed and evaluated by using, for example, QMEANDisCo Global score.

  1. The authors should report how the NMR data about the interaction-region A76-G81 were integrated into the docking of palmitoyl-L-carnitine. They should also describe the type of interactions they have observed between the substrate and region A76-G81 in the docking.

  1. An accurate analysis of the interactions between the ligand and CAC is missing for the 100 ns MD simulations.

  1. Extraction of average conformation at the initial 1 ns is questionable given that the first nanoseconds of trajectory are usually needed to the system to reach equilibrium. The authors should provide values of RMSD of the protein during the 100 ns of trajectory to prove that during 1 ns the system is already at equilibrium.

  1. On which ground the statement “ … The conformational switch from the c-state open to the m-state open facilitates the substrate to pass through the channel” is based ?

  1. The Authors should report the RMSD values for both c-state and m-state Ci and Cf conformations to convince that there are real differences in the behavior of the c-state and m-state.

  1. In the Discussion the data obtained by NMR and MD simulation should be compared.

  1. In the Discussion the data obtained in this study and those obtained in similar previous studies on mitochondrial carriers (transporters) should be compared in short.

  1. The statement “The mode of palmitoyl-L-carnitine in the c-state open and m-state open conformations of CAC represents the protein’s uptaking and release of the substrate, respectively” is not based on any experimental result.

  1. The third and final sentence of the “conclusions” does not reflect the content of the manuscript. In my opinion, the conclusions should be omitted or remarkably reworded.

Round 2

Reviewer 1 Report

No further comments.

Author Response

We sincerely thank the reviewer for the thoughtful suggestions and agreement on the publication!

Reviewer 2 Report

The revised version of the manuscript IJMS-1651531 by Ningning Zhang et al. has been significantly improved. All my concerns have been appropriately dealt with, except comment 9. The rephrased sentence "Typically, the mode of the c-state open and the m-state open conformations represents the protein's uptaking and release, respectively" is absolutely not satisfactory. It should be omitted or remarkably reworded.

Author Response

Thank the reviewer for the suggestion! We have removed the sentence.